# Tumor suppressor SMARCB1 suppresses super-enhancers to govern hESC lineage determination

Lee F Langer[1,2], James M Ward[1,3], Trevor K Archer[1]*

[1]Laboratory of Epigenetics and Stem Cell Biology, National Institute of Environmental Health Sciences, National Institutes of Health, Durham, United States; [2]Postdoctoral Research Associate Program, National Institute of General Medical Sciences, National Institutes of Health, Bethesda, United States; [3]Integrative Bioinformatics, National Institute of Environmental Health Sciences, National Institutes of Health, Durham, United States

**Abstract** The SWI/SNF complex is a critical regulator of pluripotency in human embryonic stem cells (hESCs), and individual subunits have varied and specific roles during development and in diseases. The core subunit SMARCB1 is required for early embryonic survival, and mutations can give rise to atypical teratoid/rhabdoid tumors (AT/RTs) in the pediatric central nervous system. We report that in contrast to other studied systems, SMARCB1 represses bivalent genes in hESCs and antagonizes chromatin accessibility at super-enhancers. Moreover, and consistent with its established role as a CNS tumor suppressor, we find that SMARCB1 is essential for neural induction but dispensable for mesodermal or endodermal differentiation. Mechanistically, we demonstrate that SMARCB1 is essential for hESC super-enhancer silencing in neural differentiation conditions. This genomic assessment of hESC chromatin regulation by SMARCB1 reveals a novel positive regulatory function at super-enhancers and a unique lineage-specific role in regulating hESC differentiation.

DOI: https://doi.org/10.7554/eLife.45672.001

*For correspondence: archer1@niehs.nih.gov

**Competing interests:** The authors declare that no competing interests exist.

## Introduction

Given the near complete uniformity of the DNA sequence across cell types, the regulation of chromatin accessibility, and thereby gene activity, is critical for all stages of development, including the initial fate decisions of embryonic stem cells (ESCs). Accordingly, it has been observed that widespread chromatin re-organization occurs during ESC differentiation via the activity of multiple chromatin-remodeling complexes (*Alexander et al., 2015*; *Ahn et al., 2011*; *Lessard et al., 2007*; *Dixon et al., 2015*). One of the most studied of these complexes in terms of the regulation of pluripotency and differentiation is the SWItch/Sucrose Non-Fermentable (SWI/SNF) complex. In mESCs and hESCs, the SWI/SNF complex is composed of SMARCA4 and several core subunits, including SMARCB1 (*Ho et al., 2009*; *Zhang et al., 2014*). Complex activity as a whole is essential for the full complement of pluripotency in ESCs and appears to function widely, regulating accessibility and transcription at promoters, active and poised enhancers, as well as at pluripotency factor binding sites (*Ho et al., 2009*; *Rada-Iglesias et al., 2011*; *Zhang et al., 2014*; *Alexander et al., 2015*; *King and Klose, 2017*; *Hodges et al., 2018*).

The roles of certain SWI/SNF subunits have been examined in the context of hESC pluripotency and differentiation; for example, separate studies have detailed the requirement of SMARCC2 in the maintenance of self-renewal and revealed that its negative regulation by the microRNA mir-302 is essential for efficient definitive endodermal differentiation (*Zhang et al., 2014*; *Wade et al., 2015*).

**eLife digest** Our bodies contain trillions of cells that play a wide variety of roles. Despite looking and behaving very differently to one another, all of these 'mature' cells somehow descend from a single fertilized egg that contains just one set of genes. This process is partially controlled by how 'accessible' genetic material is to the cell machinery that switches genes on or off. For example, in immature brain cells, genes required for memory are accessible, but genes needed to produce bone are not.

The developing embryo needs to control gene accessibility carefully to ensure that the right genes become available at the right time, and that crucial genes are not incorrectly 'hidden'. In humans, the protein SMARCB1 plays an important role in this process: if damaged or deleted, development will be severely disrupted, sometimes causing brain cancer early in life. However, it remains unclear how exactly SMARCB1 regulates the accessibility of its 'target' genes. Now, Langer et al. set out to answer this question, and also to determine which parts of the body need SMARCB1 to develop properly.

Human stem cells can develop into multiple mature cell types if given the right signals. Langer et al. found reducing levels of SMARCB1 prevented stem cells from maturing into brain cells, but not other kinds of cells. This suggests that SMARCB1 has a specific role in brain development, which is consistent with its devastating effect on brain health when damaged.

A detailed analysis of genetic activity and DNA accessibility showed that SMARCB1 was doing this by switching off specific regions of DNA, called stem cell super-enhancers. These regions normally enhance the activity of genes that maintain stem cells in their immature state: when certain super-enhancers are turned off by SMARCB1, this allows stem cells to progress towards a brain cell fate.

These results help us understand why damage to SMARCB1 during development causes brain cancer more often than other kinds of cancer. In the future, they could also help explain how certain types of cancer form, which would be the first step towards knowing how to treat them.

DOI: https://doi.org/10.7554/eLife.45672.002

---

However, the functions of most subunits in hESCs have not been considered, and no data are available on how they regulate chromatin accessibility in this cell type.

The core SWI/SNF subunit SMARCB1 is required for embryonic survival past implantation in mice and therefore likely has essential roles in early cell populations (*Roberts et al., 2000*). SMARCB1 is also a potent tumor suppressor, being mutated or deleted in nearly all atypical rhabdoid/teratoid tumors (AT/RTs), aggressive cancers that primarily affect the central nervous system and which can be diagnosed at very young ages, even prenatally (*Hoot et al., 2004*; *Pawel, 2018*; *Negahban et al., 2010*). Whether AT/RT-like tumors develop in mouse models is highly sensitive to the timing of SMARCB1 inactivation. Specifically, deletion at embryonic day (E)6 – E7 leads to highly penetrant CNS tumors, whereas deletion at subsequent time points has lower penetrance or no effect (*Han et al., 2016*), suggesting that a more undifferentiated state is necessary for tumorigenesis. Recent studies in SMARCB1-null cell lines have revealed that its reintroduction results in the widespread recruitment of the SWI/SNF complex to previously unoccupied enhancers, the activation of these enhancers, and the resolution of bivalency at promoters toward an active state (*Alver et al., 2017*; *Wang et al., 2017*; *Nakayama et al., 2017*). However, these findings are somewhat in disagreement as regards SMARCB1 activity at super-enhancers (*Hnisz et al., 2013*), with different groups reporting either a requirement or dispensability of SMARCB1 in targeting the SWI/SNF complex to super-enhancers and maintaining the active H3K27ac histone marks ((*Nakayama et al., 2017*; *Alver et al., 2017*; *Wang et al., 2017*; *Hnisz et al., 2013*).

We sought to explore the transcriptional and genomic impact of SMARCB1 loss in steady state and differentiating hESCs, focusing on the role of this subunit in enhancer architecture and differentiation. We find that *SMARCB1* knockdown (KD) leads to widespread transcriptional upregulation in hESCs, with an enrichment in bivalent genes, as well as differential effects on enhancer and superenhancer accessibility. Directed differentiation assays subsequently revealed that loss of SMARCB1 activity strongly inhibits neural induction in a lineage-specific manner. These findings reveal a precise

requirement for SMARCB1 in the earliest stages of development and indicate a complex, state-specific role in enhancer regulation. These results will be relevant to additional developmental stages and pathological processes, including oncogenesis.

## Results

### SMARCB1 protein reduction relieves repression of bivalent genes in hESCs

To assess the function of SMARCB1 in steady state hESCs, H1 cells were transduced with lentiviral constructs carrying a doxycycline-inducible shRNA against *SMARCB1* or a non-targeting control (*NTC*) region (*Supplementary file 1*), followed by treatment with 1 μM doxycycline (dox) for two or three days (*Meerbrey et al., 2011*; *Silva et al., 2005*). qPCR analysis and western blotting revealed strong downregulation of SMARCB1 at the transcript and protein levels, whereas dox-treated cells expressing the *NTC* control shRNA showed no such reduction (*Figure 1A*). Several previous reports have demonstrated that SWI/SNF complex stoichiometry is tightly regulated (*Chen and Archer, 2005*; *Keppler and Archer, 2010*; *Sohn et al., 2007*), raising the possibility that *SMARCB1* KD may induce instability in other complex members. Notably, *SMARCB1* KD did not decrease the protein levels of other SWI/SNF subunits, including SMARCA4, SMARCC1, SMARCC2, SMARCD1, or SMARCE1 (*Figure 1A*, *Figure 1—figure supplement 1A*). *SMARCB1* KD cells did not exhibit dramatic morphological differences from untreated controls, and no decreases in the transcript levels of *SOX2*, *OCT4*, or *NANOG* were detected, indicating a maintenance of the pluripotency transcriptional program at the analyzed time point (*Figure 1—figure supplement 1B,C*).

To assess SMARCB1 transcriptional regulatory functions, *SMARCB1* KD cells were assayed using RNAseq, with significantly affected genes being called at q > 0.05 and FC >1.5. The results showed a strong bias towards upregulation following *SMARCB1* KD (1785 up vs. 95 down) (*Figure 1B*), with the most upregulated genes including the transcription factors *ZIC1* (Fold change [FC])=13.7) and *SOX21* (FC = 13.3), and the most downregulated genes including the MYC target *LINC00176* (FC = −7.9) and the receptor *MCHR1* (FC = −3.9) (*Supplementary file 1*) (*Pérez-Morales et al., 2018*; *Tran et al., 2018*). Ingenuity Pathway Analysis (IPA) showed the top Physiological System Development and Function categories in *SMARCB1* KD cells were general developmental programs, with the most enriched category being Organismal Development (*Figure 1—figure supplement 1D*). The component functions within this category consisted of processes involved in multiple developmental processes, including neural development, angiogenesis, and genitourinary system development (*Figure 1C*). Consistent with the pathway analysis, qPCR for several early markers of the three germ layers all showed upregulation following *SMARCB1* KD, including *PAX6*, *SOX1*, *BRN2* (ectoderm), *BRACHYURY*, *GOOSECOID* (mesoderm), and *CXCR4* (endoderm) (*Figure 1D*) (*Walther and Gruss, 1991*; *Bylund et al., 2003*; *Castro et al., 2006*; *D'Amour et al., 2005*; *Ro and Dawid, 2010*; *Smith et al., 1991*). As SMARCB1 was recently reported to activate bivalent gene transcription upon reintroduction into SMARCB1$^{-/-}$ cells, we assessed the set of differentially affected genes based on their histone modification characteristics, using previously defined genes sets (*Pan et al., 2007*). Unexpectedly, we found that genes upregulated by *SMARCB1* KD were more than 50% more enriched in bivalent genes than the total considered gene set (29% vs. 18%, respectively) (*Figure 1E*). A similar percentage (28%) of the small number of downregulated genes were also bivalent (*Figure 1—figure supplement 1E*). Notably, when considering the most highly upregulated genes, the percentage of bivalent genes was nearly 3-fold that of the considered gene set (51% vs. 18%) (*Figure 1E*). Consistent with this result, we found that the TSS (±2.5 kb) of all upregulated genes significantly overlapped with the hESC ChIPseq binding sites of several members of the Polycomb Repressive Complex 2 (PRC2), including EZH2 (4.7e-12) and SUZ12 (1.0e-4), as well as the repressive histone marks H3K27me3 (1.5e-25) (*ENCODE Project Consortium, 2012*) (*Figure 1F*, *Supplementary file 1*). We also tested for enrichment for the only SWI/SNF subunit for which ChIP-seq data are available in this cell type, the catalytic subunit SMARCA4. Although these peaks did not emerge as significantly enriched near the TSS of differentially expressed genes (DEGs), this is largely due to SMARCA4 being highly biologically enriched at most promoters in hESCs. Specifically, SMARCA4 peaks are present at 16,031/17,462 (92%) of the considered genes in the RNAseq analysis, whereas these values were 90% and 88% for up- and downregulated genes, respectively

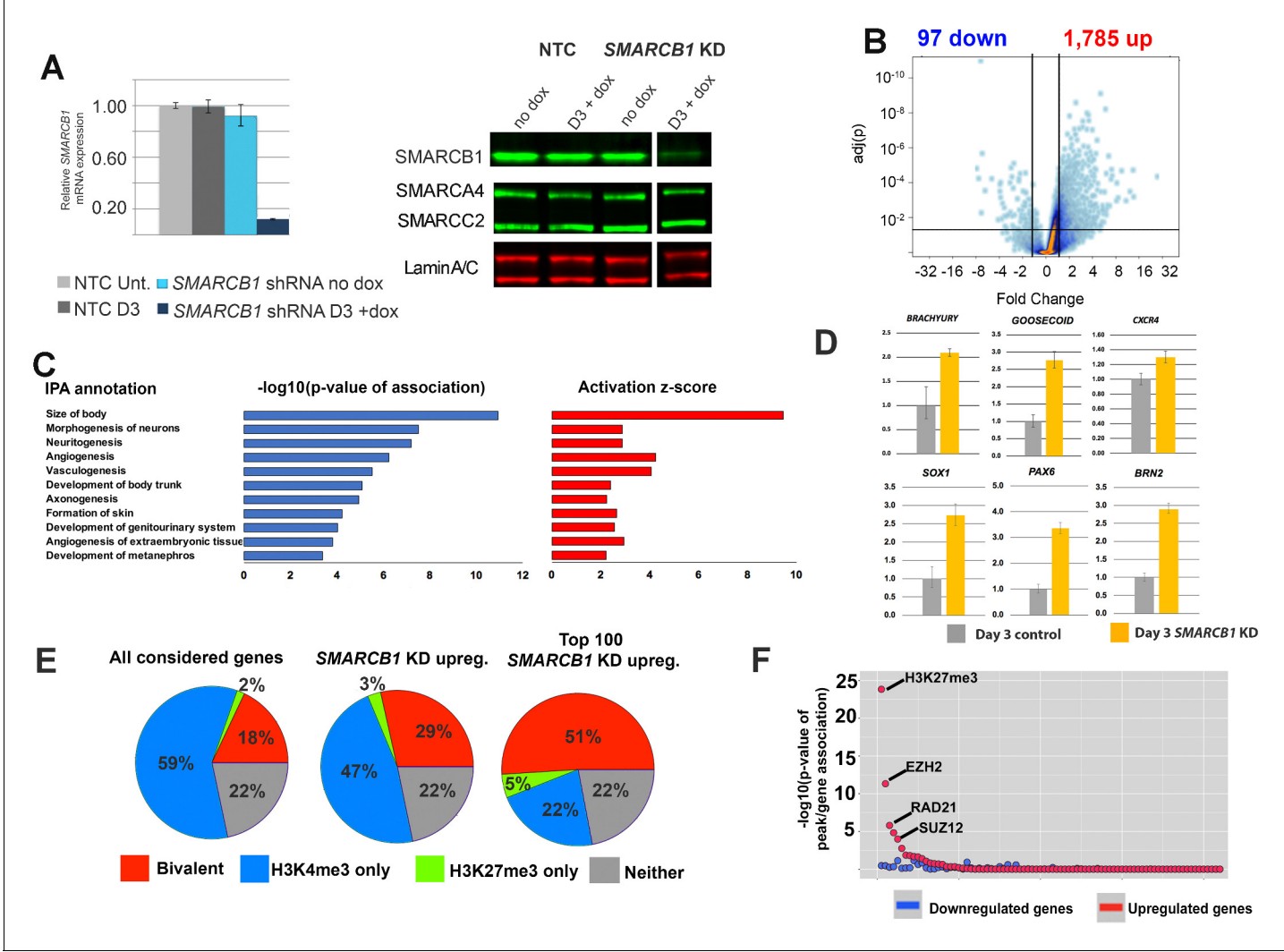

**Figure 1.** SMARCB1 is a transcriptional repressor of developmental bivalent genes. (**A**) qPCR and western blot results showing *SMARCB1* KD after 3 days (D3) of doxycycline treatment of inducible *SMARCB1* shRNA-expressing H1 hESCs. Band order/spacing was modified from the original gel. qPCR results are relative to untreated *NTC* hESCs and are normalized to the geometric mean of *18S* and *GAPDH* levels (**B**) Volcano plot showing the distribution of differentially expressed genes (q < 0.05, FC > 1.5) following 72 hr of *SMARCB1*. (**C**) IPA Organismal Development subcategories affected by *SMARCB1* KD (q < 0.05, FC > 1.5), with respective activation scores and significance values. (**D**) qPCR data showing upregulation of early markers for all three germ layers following *SMARCB1* KD. (**E**) Pie charts indicating the percentage of genes with the indicated histone marks, based on data from *Pan et al. (2007)*, in all considered genes (left), those upregulated by *SMARCB1* KD (middle), and the top 100 most upregulated genes following *SMARCB1* KD (right). (**F**) Dot plot indicating the significance of intersection between ChIPseq peaks for transcription factors and histone modifications in hESCs and the TSS (±2.5 kb) of genes significantly affected by *SMARCB1* KD. The plot is ordered with the ChIPseq peaks most significantly associated with upregulated gene TSS on the left.

DOI: https://doi.org/10.7554/eLife.45672.003

The following figure supplement is available for figure 1:

**Figure supplement 1.** Physical and transcriptional characteristics of *SMARCB1* KD hESCs.
DOI: https://doi.org/10.7554/eLife.45672.004

(*Supplementary file 1*) (*Rada-Iglesias et al., 2011*). SMARCB1 therefore appears to function largely as a transcriptional repressor in hESCs, particularly at bivalent genes, in contrast to what has been reported in experiments in mouse embryonic fibroblasts (MEFs) and several null tumor cell lines, including TTC-1240 (*Wang et al., 2017*; *Nakayama et al., 2017*; *Wilson et al., 2010*).

## *SMARCB1* KD differentially affects chromatin accessibility at key regions associated with hESC identity

To interrogate the chromatin effects of decreased SMARCB1 levels, we performed the chromatin accessibility assay ATACseq on cells subjected to *SMARCB1* KD for 48 and 72 hr, with a total of 163,782 peaks being called (*Buenrostro et al., 2013*) (*Figure 2—figure supplement 1A*). As for the RNAseq analysis, only 3day knockdown data were considered for subsequent analysis, with 15,318 peaks being lost between the untreated and knockdown conditions, and 9949 peaks being gained (*Figure 2—figure supplement 1B*). Following statistical thresholding at q < 0.05, fold change ≥1.5, *SMARCB1* KD cells were found to exhibit 4186 peaks with significantly lower accessibility and 3121 peaks with higher accessibility (Materials and methods) (*Figure 2A*). Given SMARCB1's core membership in the SWI/SNF complex, we would expect that the chromatin regions altered by loss SMARCB1 activity would significantly overlap known binding sites of the catalytic subunit SMARCA4. Indeed, both lower and higher accessibility peaks were enriched in known ChIPseq peaks for SMARCA4 (89% of lower accessibility peaks, p=2.6e-155; 77% of higher accessibility peaks, p=1.4e-12) (*Supplementary file 2*) (*Rada-Iglesias et al., 2011*).

An important readout of modified chromatin accessibility is differential gene expression, although it is difficult to assign changes in a particular ATAC peak to changes in transcriptional output. We therefore utilized an approach in which the number of differentially expressed genes (DEGs) within a given distance of all differentially accessible ATAC peaks was compared to the total number of genes within those ranges. A hypergeometric test was then used to assess whether more DEGs fell within this range than would be expected by chance. For example, when ranges of 20 kb were made around all HA peaks, 955 TSS are encompassed, 132 of which were upregulated following *SMARCB1* KD. These values respectively correspond to 5.3% of the total gene set and 7.5% of the upregulated gene set, indicating a significant enrichment of upregulated genes, with a p-value of 7.6e-5 (Also see *Figure 2—figure supplement 1C*). Plotting these significance values over ranges from 5 kB to 2 MB revealed that *SMARCB1* KD higher and lower accessibility peaks were significantly associated with up- and downregulated genes, respectively. To confirm that this association derived from the fact that these peaks were differentially accessible and was not a feature of any similar set of ATAC peaks, 1000x sets randomly selected hESC ATAC peaks were matched to the differential peaks in size and number and subjected to the same analysis. The shaded regions in *Figure 2B* correspond to the significance values for these random peaks and indicate no meaningful association with gene expression compared to differential peaks. SMARCB1 therefore both positively and negatively regulates gene expression over a range that is consistent with activity at enhancers (*Figure 2B*). Moreover, and consistent with SMARCB1's regulation of bivalent genes, the closest genes to differential peaks were significantly enriched in those classified as bivalent (*Figure 2—figure supplement 1D*).

Based on the association between differentially accessible peaks and differentially expressed genes at distances of 500 kb, (*Figure 2B*) and given previous reports of SMARCB1 activity at enhancers and super-enhancers, we assessed how *SMARCB1* KD affected accessibility of the enhancer landscape in hESCs. For this analysis, we considered hESC active and poised enhancers as well as defined human super-enhancer data sets for 98 other cell types (*Rada-Iglesias et al., 2011*; *Wang et al., 2017*; *Nakayama et al., 2017*; *Khan and Zhang, 2016*). The super-enhancers of other cell types were included in this analysis given previous results indicating that manipulation of the SWI/SNF complex can promote or repress particular fates (*Wade et al., 2015*; *Zhang et al., 2014*). As super-enhancers are key regions associated with cell identity, the preferential localization of differential ATAC peaks in these regions for other cell types would inform any effects of *SMARCB1* KD on differentiation. *SMARCB1* KD lower accessibility peaks were enriched in active hESC enhancers, such as an enhancer located 12 kB upstream of *KCNQ2*, as well as poised hESC enhancers, with limited enrichment in other enhancer sets (*Figure 2C and D*, *Supplementary file 2*) (*Rada-Iglesias et al., 2011*). In agreement with this result, enrichment analysis using publicly available hESC ChIPseq peaks for histone marks showed strong enrichment for *SMARCB1* KD lower accessibility peaks in the enhancer markers H3K4me1 (p=2.1e-218), H3K27ac (p=4.4e-36), and H3K4me2 (p=1.2 e −153) (*Figure 2—figure supplement 1E*, *Supplementary file 2*) (*ENCODE Project Consortium, 2012*). These results are consistent with the previously reported role of SMARCB1 in maintaining enhancer accessibility (*Wang et al., 2017*; *Nakayama et al., 2017*). Of all considered enhancers sets, *SMARCB1* KD higher accessibility peaks were enriched only in H1 hESC super-

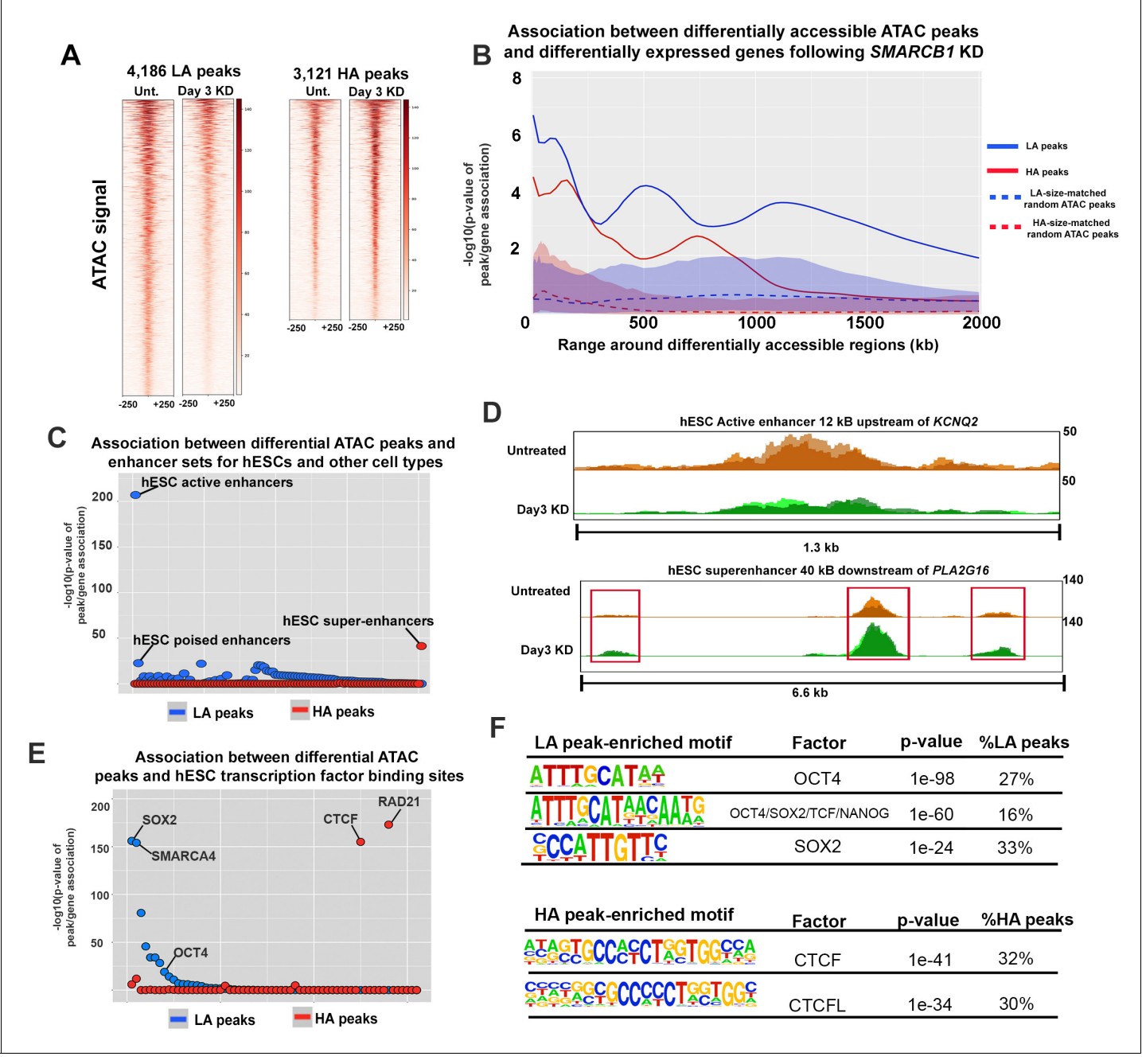

**Figure 2.** SMARCB1 negatively regulates accessibility at key pluripotency regions in hESCs. (A) Heatmaps showing the ATAC signal over the peaks with lower (LA) and higher (HA) accessibility (q < 0.05, FC > 1.5) prior to (Unt.) and following 72 hr of *SMARCB1* KD. (B) Plot of the significance of the association between *SMARCB1* KD lower/higher accessibility peaks (solid lines) and down/up-regulated genes, respectively, by RNAseq (q < 0.05, FC > 1.5) over 2 Mb kb. The blue/red-shaded regions reflect the 5–95% confidence interval (CI) for the significance of the association between down/up-regulated genes and 1000x sets of randomly selected hESC ATAC peaks that were matched in size and number to the lower/higher accessibility ATAC peak sets. The dotted lines indicate the median of the random peak set-based significance range. (C) Dot plot of the significance of the association between *SMARCB1* KD lower/higher accessibility peaks and human enhancer regions. The plot is ordered with the enhancer regions most significantly associated with lower accessibility peaks on the left. (D) Top: ATAC signal tracks for untreated and 72 hr *SMARCB1* KD cells over an active hESC enhancer 12 kB upstream of *KCNQ2*. Bottom: ATAC signal tracks for untreated and 72 hr *SMARCB1* KD cells over an hESC super-enhancer with three higher accessibility peaks, 40 kB downstream of *PLA2G16*. (E) Dot plots indicating the significance of intersection between *SMARCB1* KD lower/higher accessibility peaks and hESC transcription factor binding sites. The plot is ordered with the ChIPseq peaks most significantly associated with lower accessibility peaks on the left. (F) Top: Pluripotency factor related motifs that are significantly enriched in lower accessibility peaks following *SMARCB1* KD, as well as the significance of the association and the percentage of these peaks that contain the motif. Bottom: The significance of the

*Figure 2 continued on next page*

*Figure 2 continued*

association and the percentage of higher accessibility peaks that contain CTCF and CTCFL motifs following *SMARCB1* KD, as well as the significance of the association and the percentage of these peaks that contain the motif.

DOI: https://doi.org/10.7554/eLife.45672.005

The following figure supplement is available for figure 2:

**Figure supplement 1.** Chromatin accessibility characteristics of *SMARCB1* KD hESCs.

DOI: https://doi.org/10.7554/eLife.45672.006

enhancers (p=4.6e-42), being present in ~19% (127/684) of these regions (*Hnisz et al., 2013*) (*Figure 2C*). A subset of these super-enhancers (34/127, 27%) contained multiple higher accessibility peaks, including the super-enhancer 40 kB downstream of the phospholipase *PLA2G16*, indicating that some super-enhancers may be more strongly regulated by SMARCB1 than others (*Figure 2D*, *Figure 2—figure supplement 1F*, *Supplementary file 2*). These data are the first to show that SMARCB1 negatively regulates super-enhancers in any cell type and indicate that this subunit differentially regulates accessibility across the hESC enhancer landscape.

To assess how the loss of SMARCB1 affects accessibility at smaller-scale hESC regulatory regions, we analyzed the distribution of higher and lower peaks in terms of all ENCODE ChIPseq datasets for hESCs as well as a previously published SOX2 ChIPseq dataset (*Zhou et al., 2016*; *ENCODE Project Consortium, 2012*). In addition to SMARCA4 binding sites, *SMARCB1* KD lower accessibility peaks were significantly enriched in SOX2 (p=1.8E-156) and OCT4 (p=8.0e-15) ChIPseq peaks, indicating that SMARCB1 is necessary to maintain chromatin accessibility at key pluripotency factor binding sites (*Figure 2E*, *Supplementary file 2*). In contrast, *SMARCB1* KD higher accessibility peaks were enriched in binding sites for the cohesin complex member RAD21 (p=7.3E-174) and the cohesin-interacting protein CTCF (p=2.3E-155). In fact, 50% of *SMARCB1* KD HA peaks overlapping hESC RAD21 and CTCF binding sites, consistent with the enrichment of these sites near the TSS of upregulated genes (*Figure 1F*, *Figure 2E*, *Supplementary file 1*, *Supplementary file 2*) (*Parelho et al., 2008*). In agreement with the ChIPseq peak enrichment findings, motif analysis showed that lower accessibility peaks were enriched in OCT4 and SOX2 binding motifs, whereas higher accessibility peaks were associated with CTCF and CTCF-like binding motifs (*Figure 2F*, *Supplementary file 2*).

These data indicate reveal a complex picture of chromatin regulation by SMARCB1 at pluripotency-associated regions and are the first to reveal a negative regulatory role at super-enhancers, a significant result in terms of this subunit's role in hESC differentiation.

## SMARCB1 is required for the initial stages of neural induction but is dispensable for endodermal and mesodermal induction

We next performed directed differentiation assays to assess how the loss of SMARCB1 activity would affect hESC differentiation down the three germ lineages. We found that *SMARCB1* KD cells successfully executed definitive endodermal differentiation, based on the expression of the transcription factor SOX17 and the surface marker CXCR4 (*Figure 3A*, *Figure 3—figure supplement 1A–C*) (*D'Amour et al., 2005*; *Liu et al., 2007*). Mesodermal induction was similarly unimpaired, with *SMARCB1* KD cells expressing the early mesodermal marker EOMES/TBR2 as well as several other markers, including *HAND1*, *GOOSECOID*, and *FOXF1* (*Russ et al., 2000*; *Barnes et al., 2010*; *Niehrs et al., 1994*) (*Figure 3B*). In contrast to the results for endodermal and mesodermal induction, *SMARCB1* KD cells exhibited a robust resistance to neural induction. Control cultures exhibited large areas of PAX6$^+$/OCT4$^-$ cells, a robust upregulation of *PAX6* transcript levels and increased expression of several neural stem cell markers (*Figure 3C*). In striking contrast, *SMARCB1* KD cells showed few PAX6$^+$ cells, numerous strongly positive OCT4$^+$ cells, and a greatly attenuated upregulation of *PAX6* transcript levels. The upregulation of other NSC markers was similarly blunted (*Figure 3C*). Similar results were obtained in a second *SMARCB1* KD line, while NTC-shRNA-expressing cells showed no such resistance to induction on dox treatment (*Figure 3—figure supplement 1D*). To test the robustness of this resistance of *SMARCB1* KD cells to neural induction, we generated embryoid bodies (EBs) from control and *SMARCB1* KD cells and subjected them to a '4-/4+' protocol consisting of four days of growth without retinoic acid (RA) and 4 days with 1 µM RA (*Bain et al., 1995*). While control EBs were large and globular, as has previously been observed in

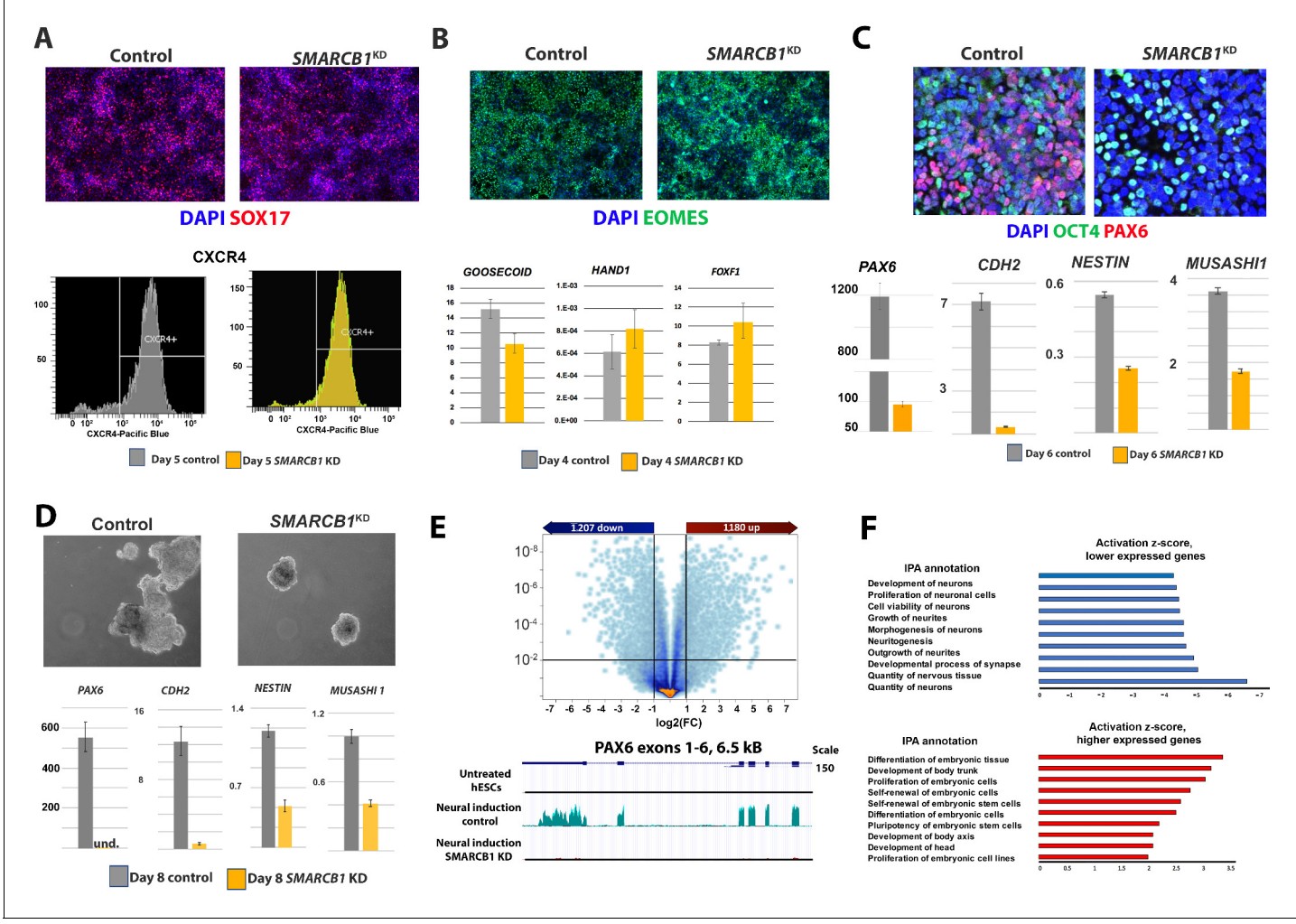

**Figure 3.** *SMARCB1* KD prevents neural induction and upregulation of neural differentiation-related genes. (**A**) Top: Control and *SMARCB1* KD cells subjected to a 5 day definitive endoderm protocol induction protocol stained for SOX17 (red) and with DAPI (blue). Bottom: Flow cytometry histograms showing similar percentages of CXCR4[+] cells in control and *SMARCB1* KD cells subjected to this protocol. (**B**) Top: Control and *SMARCB1* KD cells subjected to a 4-day mesodermal induction protocol stained for EOMES (green) and with DAPI (blue). Bottom: qPCR analysis of control and *SMARCB1* KD cells subjected to the same protocol for the early mesodermal markers *GOOSECOID*, *HAND1*, and *FOXF1*. All qPCR results are relative to steady state hESCs and are normalized to the geometric mean of *18S* and *GAPDH* levels. (**C**) Top: Control and *SMARCB1* KD cells subjected to a 6 day directed neural induction protocol stained for PAX6 (red), OCT4 (green), and with DAPI (blue). Bottom: qPCR analysis showing a failure of *SMARCB1* KD cells to upregulate neural differentiation markers *PAX6*, *CHD2*, *NESTIN*, and *MS1*. (**D**) Top: Control and *SMARCB1* KD embryoid bodies (EBs) subjected to a neural induction protocol. Bottom: qPCR analysis showing a failure of *SMARCB1* KD cells to upregulate the neural differentiation markers *PAX6*, *CHD2*, *NESTIN*, and *MS1*. (**E**) Top: Volcano plot illustrating the extent and significance of differential gene expression between control and *SMARCB1* KD cells subjected to neural induction protocol as determined by RNAseq (q < 0.01 and FC > 2.0). Bottom: RNAseq tracks *PAX6* for steady state hESCs as well as control and *SMARCB1* KD cells subjected to a monolayer neural induction protocol. Flat lines indicate undetectable or nearly undetectable expression at the scale used. (**F**) Top: Activation scores for IPA *Nervous System Development* sub-categories, considering genes expressed at lower levels in *SMARCB1* KD cells compared to controls following the neural induction protocol. Bottom: Activation scores for IPA *Embryonic Development* sub-categories, considering genes expressed at higher levels in levels in *SMARCB1* KD cells compared to controls following the neural induction protocol.

DOI: https://doi.org/10.7554/eLife.45672.007

The following figure supplement is available for figure 3:

**Figure supplement 1.** Flow cytometry gating strategy and qPCR data for negative control as well as 2[nd] *SMARCB1* KD line control.

DOI: https://doi.org/10.7554/eLife.45672.008

neural induction protocols, *SMARCB1* KD EBs remained small and spherical (*Stanslowsky et al., 2016*). Moreover, whereas several neural differentiation-related genes, including *PAX6*, were highly upregulated in the control condition, *SMARCB1* KD EBs exhibited undetectable *PAX6* levels and either attenuated upregulation or downregulation of other NSC markers compared to steady state conditions (*Figure 3D*).

To obtain a more comprehensive picture of the transcriptional effects of *SMARCB1* KD in cells subjected to neural induction, we performed RNAseq on control and *SMARCB1* KD cells subjected to the monolayer neural induction protocol. We identified 1207 genes that were more highly expressed in the *SMARCB1* KD condition and 1180 genes with lower expression (q < 0.01, FC > 2.0) (*Figure 3E*). When the statistical constraints were relaxed (q < 0.05, FC > 1.5), the numbers of higher and lower expressed genes more than doubled to 2677 and 3,638, respectively. Some of the genes with the highest expression in *SMARCB1* KD vs. control cells included the pluripotency markers *TDGF1* (Fold difference [FD]=35.9), *FGF4* (FD = 26.8) and *NANOG* (FD = 23.5). In contrast, some of the most highly repressed genes in *SMARCB1* KD vs. control cells included the neural differentiation markers *FOXG1* (FD = −28.2), *DLK* (FD = −25.9), and LHX2 (FD = −14.2) (*Supplementary file 3*) (*Watanabe et al., 2005*; *Porter et al., 1997*; *Tedeschi and Bradke, 2013*). Moreover, consistent with the qPCR results, the RNAseq data confirmed that *PAX6* was expressed at appreciable levels only in control cells (*Figure 3E*). As expected, IPA analysis revealed that the genes expressed at lower levels in *SMARCB1* KD cells were enriched in several categories related to neural development. In contrast, more highly expressed genes were enriched in categories related to the self-renewal of stem cells and the maintenance of pluripotency (*Figure 3F*).

Together, these results show that *SMARCB1* KD cells have a lineage-specific requirement for neural induction and that loss of SMARCB1 activity results in both a failure to upregulate neural genes as well as a failure to downregulate pluripotency-related pathways.

## SMARCB1 is required for increased accessibility of chromatin regions associated with neural differentiation

To address mechanisms by which *SMARCB1* KD prevents neural induction, we subjected control and *SMARCB1* KD cells to ATACseq midway through (Day 3) and at the completion of the neural induction protocol (Day 6). Here, we focus on differences between the control and *SMARCB1* KD cells at Day 6, although regions of interest were defined using data from all starting, Day 3, and Day six conditions. Of the considered 88,749 ATAC peaks, 7801 exhibited higher accessibility (HA) in *SMARCB1* KD cells, whereas 13,614 exhibited lower accessibility (LA) (q < 0.01, FC >2.0) (*Figure 4A*, *Figure 4—figure supplement 1A*, Materials and methods).

To assess the effects of *SMARCB1* KD on the accessibility of regions that normally exhibit open chromatin in neural stem cells (NSCs), we compared differential peaks with over 80,000 previously published regions known to be accessible in human cortical NSCs but not in pluripotent stem cells (*Forrest et al., 2017*). We found that 30% (4,107/13,614) of lower accessibility peaks overlapped NSC-specific regions (p<2.2E-308) (*Figure 4—figure supplement 1B*). No such enrichment was observed for higher accessibility peaks, with only 555/7,801 (7%, p=1) overlapping NSC-specific accessible regions (*Figure 4—figure supplement 1B*). The accessibility signal over these 4107 lower accessibility peaks in *SMARCB1* KD cells following the neural induction protocol was virtually indistinguishable from the signal for hESCs, indicating that the increase in accessibility normally observed in these regions during neural induction was completely abrogated (*Figure 4B*). Moreover, several genes important for neural differentiation had multiple nearby regions that normally gain accessibility during differentiation but that failed to do so in *SMARCB1* KD cells. For example, within the multiple ATAC-sensitive peaks in and around the PAX6 locus, one can clearly identify six such regions were located upstream and down of the *PAX6* locus that are substantially depressed (*Figure 4C*).

We next asked whether these lower accessibility peaks were associated with differentially expressed genes as determined by RNAseq. The results indicated that lower accessibility peaks were significantly associated with differential gene expression to nearly 1 Mb (*Figure 4D*), after which a significant effect was not observed. Importantly, an analysis of 1000x sets of randomly selected size-matched ATAC peaks showed no such association (*Figure 4D*). This association between lower accessibility peaks and gene expression are suggestive of the activity of distal enhancers. Moreover, several promoters of neural/neural differentiation-related genes contained lower accessibility peaks (e.g., *CER1, CRX, DPP6, SIX1*), while multiple promoters of genes related

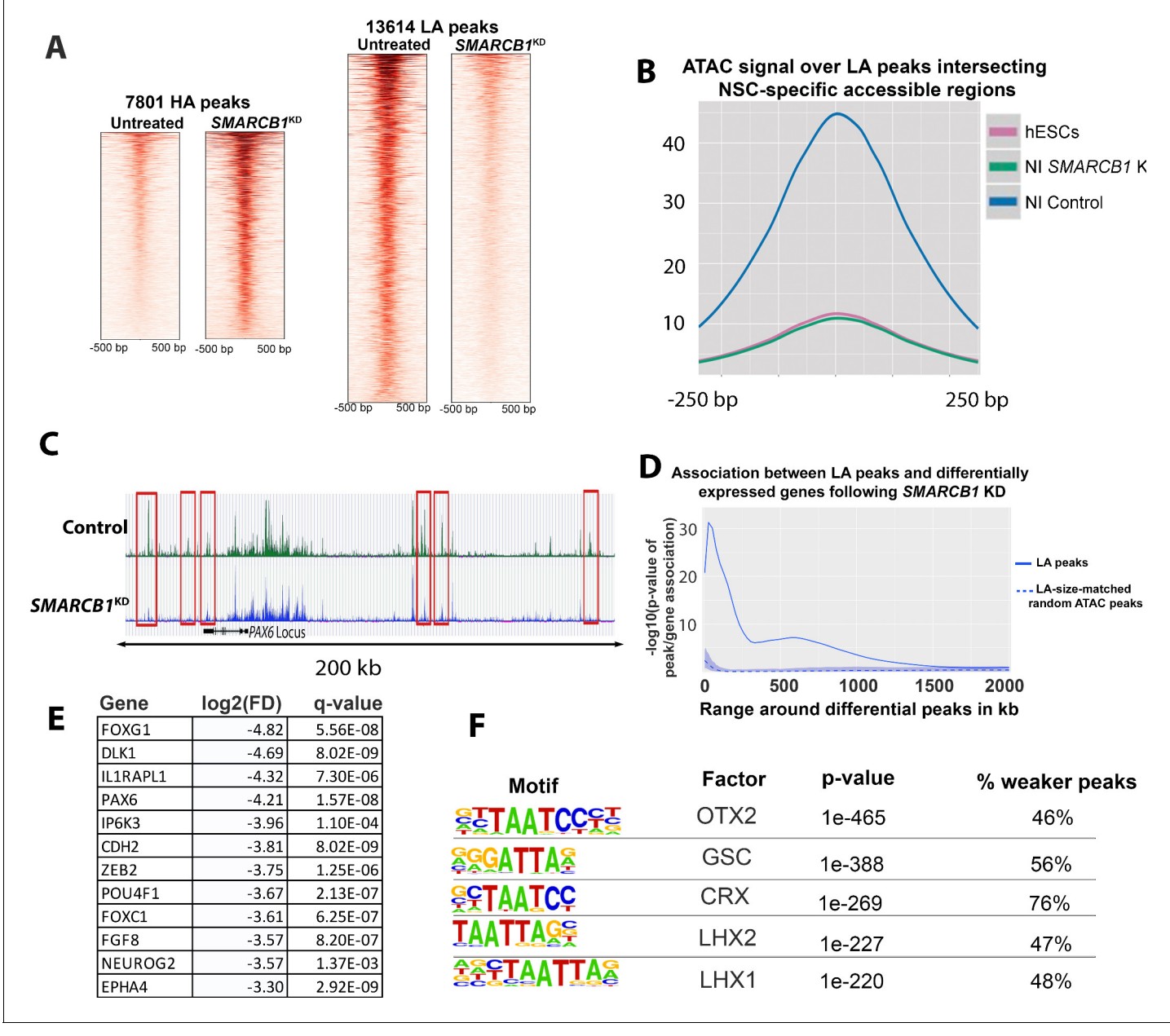

**Figure 4.** *SMARCB1* KD disrupts accessibility dynamics near neural differentiation-related genes. (A) Heatmaps showing the normalized ATAC signal for control and *SMARCB1* KD cells in 7801 peaks with significantly (q < 0.01, FC >2.0) higher (HA) and 13,614 peaks with lower accessibility (LA) peaks following *SMARCB1* KD. (B) Accessibility signal over 4107 peaks that are uniquely accessible in NSCs but not iPSCs and which exhibit lower accessibility in *SMARCB1* KD cells (*Forrest et al., 2017*). (C) Accessibility track over the *PAX6* locus for steady state hESCs as well as control and *SMARCB1* KD cells subjected to a monolayer neural induction protocol. NSC-specific peaks with significantly lower accessibility than the control condition are indicated with red rectangles. (D) Solid line: Significance of the association between lower accessibility peaks and differentially expressed genes (q < 0.01, FD >2.0) over a 2 Mb range. The blue shaded region reflects the 5–95% confidence interval (CI) for the significance of the association between differentially expressed genes and 1000x sets of randomly selected ATAC peaks that were matched in size and number to the lower accessibility ATAC peak set. The dotted line indicates the median of the random peak set-based significance range. (E) Selected differentially expressed genes between the *SMARCB1* KD and control condition that have ≥1 lower accessibility region within 500 kb and which are in the IPA *Neuron Development* Pathway. (F) Motif analysis of the peaks that show stronger or weaker accessibility in *SMARCB1* KD cells subjected to the neural induction protocol compared to control cells.

DOI: https://doi.org/10.7554/eLife.45672.009

The following figure supplement is available for figure 4:

**Figure supplement 1.** Chromatin accessibility characteristics of SMARCB1 KD cells subjected to neural induction protocol.

*Figure 4 continued on next page*

*Figure 4 continued*

DOI: https://doi.org/10.7554/eLife.45672.010

to pluripotency contained higher accessibility peaks (e.g., including *OCT4*, *mir302*, *DNMT3B*, *ZIC3*, and *DPPA2/4*) (*Supplementary file 4*).

IPA analysis showed that differentially affected genes within 500 kb of lower accessibility peaks were enriched in genes within the *Neuron Development* pathway, for which a strongly negative activation score was obtained (p=2.6E-40, z = −2.935, *Supplementary file 4*). Many of these genes are regulators of neural differentiation and are early markers of NSCs, including *PAX6* and *FOXG1* (*Figure 4E*, *Supplementary file 4*)(*Walther and Gruss, 1991*; *Watanabe et al., 2005*). Consistent with the above data, HOMER motif analysis of lower accessibility peaks revealed enrichment in motifs for transcription factors that regulate neural development, including *Otx2*, *Lhx1*, and *Lhx2* (*Figure 4F*, *Figure 4—figure supplement 1D*) (*Ang et al., 1996*; *Porter et al., 1997*; *Shawlot and Behringer, 1995*). Together, these data indicate that SMARCB1 is required for the positive changes in chromatin accessibility, and thereby the necessary increases in gene expression, that are required during the early stages of neural induction.

## SMARCB1 KD prevents silencing of hESC super-enhancers

We next determined whether *SMARCB1* KD cells maintained hESC chromatin characteristics, including accessibility at enhancers, super-enhancers, and pluripotency factor binding sites during the initial stages of neural induction. We found enrichment of these powerful regulatory stretches among higher accessibility peaks, with 282 (41%) hESC super-enhancers intersecting at least one higher accessibility peak (p=5.8E-65) (*Figure 5A*, *Supplementary file 5*). In contrast to what was observed in the steady state, active hESC enhancers were also enriched in higher accessibility peaks, with 562 higher accessibility peaks intersecting this set (p=1.2E-35, *Figure 5A*) (*Rada-Iglesias et al., 2011*). Importantly, no enrichment in enhancers or super-enhancers was observed for lower accessibility peaks (both p=1) (*Figure 5—figure supplement 1*). The maintenance of accessibility over hESCs super-enhancers was even more evident when the normalized accessibility signal was used to generate metaplots over these regions. Whereas the accessibility signal for control cells exhibited a ≈30% decrease relative to steady state hESCs, the accessibility signal of *SMARCB1* KD cells was essentially unchanged from hESCs not subjected to the induction protocol (*Figure 5B*). An example of this was observed with the super-enhancer near the kinase *DAPK1*, which contains three higher accessibility regions with comparable accessibility to what is seen in steady state hESCs (*Figure 2B*). This result indicates that the accessibility of hESC super-enhancers was strongly maintained throughout the neural induction protocol. Notably, there was no enrichment among higher accessibility peaks for any of the 98 other analyzed super-enhancer sets for different cell types (*Supplementary file 5*) (*Khan and Zhang, 2016*).

As super-enhancers are drivers of cell identity and have a strong effect on gene expression, we calculated the association between hESC super-enhancers with higher accessibility peaks and differentially expressed genes by RNAseq (*Hnisz et al., 2013*). We found that these higher accessibility peaks were correlated with differential gene expression out to 1 Mb (*Figure 5C*). The strongest association was seen within 500 kB of higher accessibility peaks within super-enhancers, and a slowly declining association was observed for more distal genes. As before, 1000x sets of randomly selected sized-matched ATAC peaks did not show a significant association with differential gene expression (*Figure 5C*). As expected based on the previously described localization of super-enhancers near genes related to cell identity, several of the differentially expressed genes near HA-super-enhancer regions were found to be powerful positive regulators of pluripotency, including *OCT4* and the microRNA *mir-302* (*Figure 5C*) (*Hnisz et al., 2013*). To assess whether these enhancers exhibited maintained activity, we analyzed levels of transcription in these regions. Consistent with maintained activity, higher levels of transcription were detected in 89 hESC super-enhancers in *SMARCB1* KD cells compared to the control condition, whereas lower levels of transcription were observed at only 23 super-enhancers (*Figure 5D*, *Supplementary file 5*). Moreover, of the 89 with higher eRNA expression, 58 (65%) had closest genes that were differentially

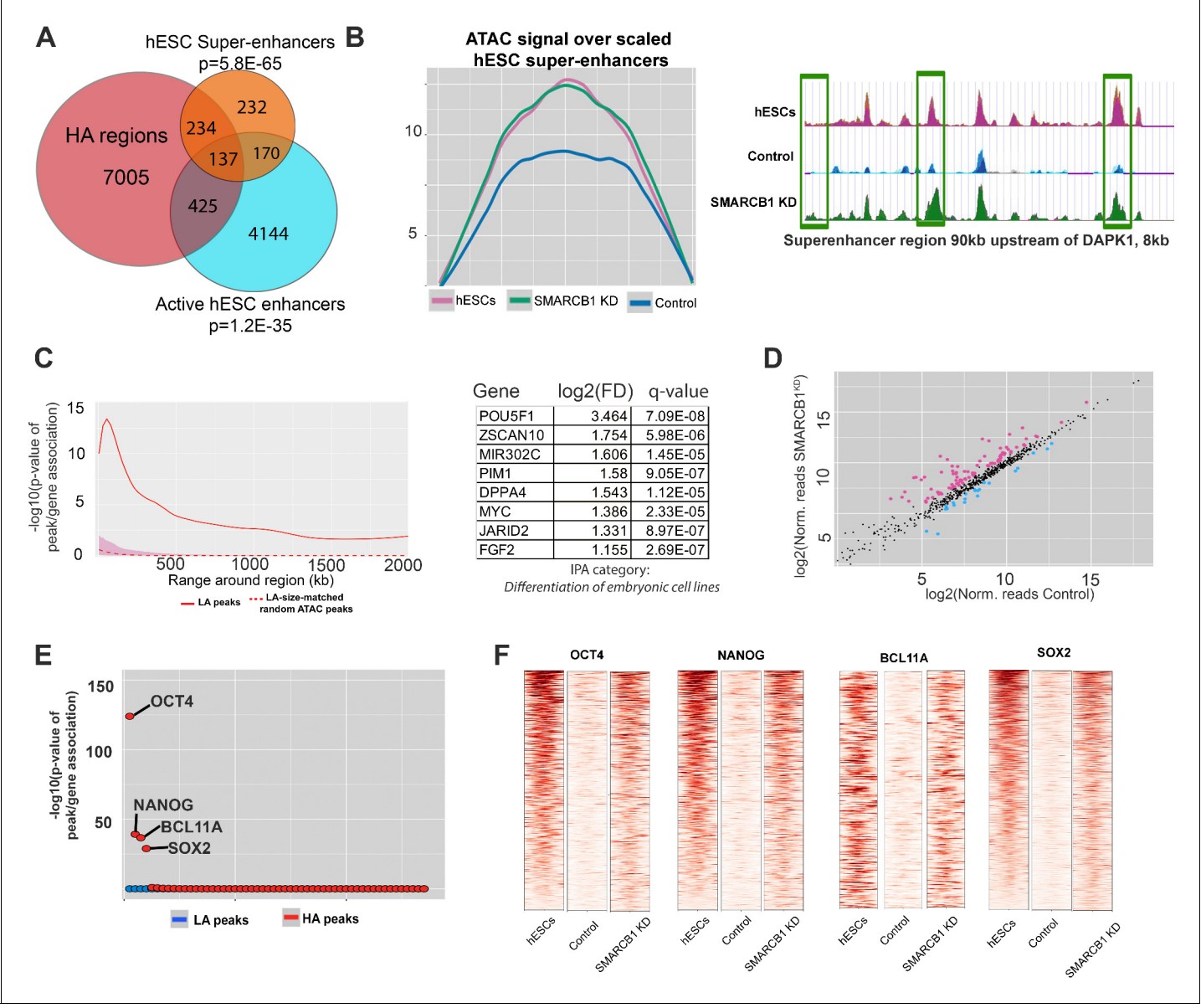

**Figure 5.** hESC chromatin landscape resists silencing during neural induction protocol in *SMARCB1* KD cells. (**A**) Venn diagram showing the overlap between higher accessibility peaks in *SMARCB1* KD cells subjected to the neural induction protocol and hESC enhancers and super-enhancers. The degree of significance of the overlap is given. (**B**) Left: Accessibility signals over hESC super-enhancers for steady state hESCs as well as control and *SMARCB1* KD cells subjected to neural induction protocol. Right: Track showing normalized accessibility signals for steady state hESCs as well as control and *SMARCB1* ^KD^ cells subjected to neural induction protocol over an 8 kb portion of a SE near the *DAPK1* locus. Differentially accessible peaks are marked with green boxes. (**C**) Left: Significance of the association between peaks with higher accessibility in hESC super-enhancers and differentially expressed genes by RNAseq over a range of 2 Mb. The red shaded region reflects the 5–95% confidence interval (CI) for the significance of the association between differentially expressed genes and 1000x sets of randomly selected ATAC peaks that were matched in size and number to the set of higher accessibility ATAC peaks in hESC super-enhancers. The dotted line indicates the median of the random peak set-based significance range. Right: Differentially affected genes within 500 kB of an hESC SE with at least one higher accessibility region in *SMARCB1* KD cells following the neural induction protocol. (**D**) Scatterplot showing the number of normalized RNAseq reads over hESC super-enhancers in both control and *SMARCB1* KD cells subjected to a neural induction protocol. Pink dots indicate regions with significantly higher levels of RNA, and blue dots indicate regions with lower levels of transcription. (**E**) Dot plots indicating the significance of intersection between *SMARCB1* KD lower/higher accessibility in the neural induction experiments and hESC transcription factor binding sites. The plot is ordered with the ChIPseq peaks most significantly associated with higher accessibility on the left. (**F**) Heat map indicating the significance of overlap between hESC ChIPseq peaks and higher/lower accessibility in *SMARCB1* KD cells following the neural induction protocol. Darker colors indicate a higher degree of significance. Right: Normalized accessibility signals over hESC binding sites for OCT4, NANOG, BCL11A, and SOX2 for steady state hESCs as well as control and *SMARCB1* KD cells subjected to the neural induction protocol.

*Figure 5 continued on next page*

*Figure 5 continued*

DOI: https://doi.org/10.7554/eLife.45672.011

The following figure supplement is available for figure 5:

**Figure supplement 1.** Intersection of LA peaks with hESC enhancer sets.

DOI: https://doi.org/10.7554/eLife.45672.012

expressed by RNAseq, most of which (51/58, 88%) were expressed at higher levels in *SMARCB1* KD cells (*Supplementary file 5*).

To assess whether smaller-scale features of the hESCs chromatin landscape might also be maintained in *SMARCB1* KD cells, we evaluated the accessibility of known hESC pluripotency factor binding sites. Indeed, higher accessibility peaks were enriched in the binding sites for several key hESC TFs, including OCT4 (p=5.3e-125), NANOG (p=6.7e-40), and SOX2 (p=1.2e-29) (*Figure 5E*, *Supplementary file 5*). This overlap comprised a significant percentage of these binding sites, including 14%, 10% and 7% of hESC SOX2, OCT4, and NANOG, binding sites, respectively. Moreover, higher accessibility peaks overlapped 14% of hESC peaks for the SWI/SNF-associated factor BCL11A (p=2.1e-37), suggesting that the SWI/SNF complex mediates the repression of these regions during the early stages of neural induction in a SMARCB1-dependent manner (*Figure 5E*, *Supplementary file 5*). In all of these pluripotency factor binding sites, the normalized accessibility signal in *SMARCB1* KD cells subjected to neural induction was similar to that of untreated hESCs, whereas the signal for control cells was strongly attenuated compared to the hESC accessibility signal, indicating a failure to repress accessibility in these regulatory regions in *SMARCB1* KD conditions (*Figure 5F*).

## Discussion

Several SWI/SNF subunits are known to be essential for hESC to maintain their full complement of pluripotency (*Zhang et al., 2014*; *Ho et al., 2009*; *Schaniel et al., 2009*). However, there are few data on the specific role of core subunits in genome-wide chromatin accessibility in hESCs, nor has the role of SMARCB1 been explored in terms of its role in regulating hESC differentiation. An inducible knockdown strategy permitted the assessment of SMARCB1's regulation of transcription and chromatin accessibility in steady state hESCs and under differentiation conditions. These data revealed surprising insights into SMARCB1's regulation of the hESC enhancer landscape and its specific requirement for neural induction.

The observed widespread upregulation of bivalent genes in *SMARCB1* KD hESCs was unexpected given previous reports that SMARCB1 loss leads to PCR2-mediated repression in MEFs and that reintroduction leads to bivalent gene activation in *SMARCB1*-null cell lines (*Nakayama et al., 2017*; *Wilson et al., 2010*). Given the highly euchromatic nature of embryonic stem cells and the developmental repercussions of premature differentiation, it is possible that the SWI/SNF complex has a more repressive role in hESCs than in differentiated or tumor lines (*Meshorer and Misteli, 2006*). In line with this interpretation, a previous microarray analysis of *SMARCA4* KD in hESCs revealed a bias in gene upregulation (472/529, 74%), a result that is consistent with our unpublished observations (*Zhang et al., 2014*). It is also worth noting that while there were a greater number of lower accessibility peaks in the steady *SMARCB1* KD condition, there was a strong bias towards transcriptional upregulation. We hypothesize that SMARCB1 positively affects transcription by other mechanisms than altered accessibility. For example, SMARCB1 may be required for the recruitment of transcription factors or transcriptional machinery to promoters hESCs, which would result in decreased transcription but not a dramatic change in accessibility. Evidence for transcriptional control independent of accessibility changes include the established antagonism between the SWI/SNF and PRC2 complexes, the interaction between the complex and the tumor suppressor p53, and the observed associations between SMARCB1 and RNA Pol I and RNA Pol II (*Kadoch et al., 2017*; *Lee et al., 2002*; *Cho et al., 1998*; *Zhai et al., 2012*).

The role of SMARCB1 at enhancers has received significant attention in recent years, and the results are not wholly in agreement. Specifically, it was reported that SMARCB1 deletion in MEFs decreases levels of the active markers H3K27ac and H3Kme1 at enhancers, whereas super-enhancers

were relatively spared (*Wang et al., 2017*). In contrast, others have found that both enhancers and super-enhancers show increased levels of H3K27ac upon SMARCB1 reintroduction into null tumor cell lines (*Nakayama et al., 2017*). Our data are consistent with both sets of previous results in that *SMARCB1* KD in steady state hESCs leads to widespread loss of enhancer accessibility. However, we find that SMARCB1 has a repressive role at hESC super-enhancers, a previously undescribed effect and one not observed for any of the 98 other analyzed human super-enhancer datasets. This is a salient difference in the context of AT/RT tumorigenesis in that the cell of origin is likely an undifferentiated NSC, indicating that current models of SMARCB1 activity at enhancers that are based on differentiated cells or reintroduction experiments may not fully capture the functions that contribute its tumorigenic role (*Han et al., 2016*).

That *SMARCB1* KD leads to elevated accessibility at CTCF binding sites is particularly noteworthy given recent findings that a SMARCB1-excluding non-canonical SWI/SNF complex (ncBAF, also termed GBAF) preferentially targets CTCF sites in mESCs and several tumor cell lines (*Gatchalian et al., 2018*; *Michel et al., 2018*). That CTCF sites gain accessibility in *SMARCB1* KD conditions suggests that an absence of SMARCB1 may promote the formation of the ncBAF. This is an intriguing possibility given that the ncBAF complex positively regulates naïve pluripotency in mESCs and may provide a mechanism by which SMARCB1-deficient cells resist differentiation (*Gatchalian et al., 2018*).

The lineage specificity of SMARCB1's requirement in neural differentiation is relevant in the context of its role as a tumor suppressor. Although loss of SMARCB1 in adult mice leads to lymphoma, SMARCB1 mutation-associated pediatric AT/RTs are found in the CNS, a finding that has been recapitulated in a conditional *SMARCB1* KO mouse model (*Han et al., 2016*; *Roberts et al., 2000*; *Babgi et al., 2018*). Neural differentiation has been reported to be the default lineage choice of ESCs, and recent work indicates that this characteristic is cell-intrinsic and driven in part by expression of the zinc finger protein Zfp521 (*Muñoz-Sanjuán and Brivanlou, 2002*; *Kamiya et al., 2011*). It is possible that loss of SMARCB1 activity renders cells deficient in the mechanisms involved in this intrinsic process, leaving differentiation pathways that normally require extrinsic stimuli unimpaired.

The above results provide critical insights into how a core SWI/SNF subunit regulates both the hESC enhancer landscape as well as differentiation down a lineage where it is strongly implicated as a developmental tumor suppressor. The complex roles that SMARCB1 plays at different enhancer subtypes should be borne in mind when assessing both subsequent stages of development as well as the initial stages of cellular transformation.

## Materials and methods

### Cell culture

H1 hESCs were cultured in *TeSR-E8* growth medium (STEMCELL Technologies, #05990) at 37°C in a 5% $CO_2$ atmosphere. Cells were grown feeder-free on a substrate of hESC-qualified Matrigel Matrix (Corning #354277). Cells were passaged at or before 80% confluency and at densities of 1:6-1:24, unless otherwise specified.

Generation and use of doxycycline-inducible KD lines shRNAs were a kind gift from the laboratory of Dr. Guang Hu and were described in *Silva et al. (2005)*. Sub-cloning of the non-targeting shRNA and those against *SMARCB1* and *SMARCA4* were performed by digestion of the pINDUCER backbone vector with XhoI (New England BioLabs, #R0146) and MluI (New England BioLabs, #R0198), following standard procedures (*Meerbrey et al., 2011*). Lentiviruses carrying the respective shRNAs were produced at the NIEHS Viral Vector Core Laboratory according to a previously established protocol (*Salmon and Trono, 2007*). H1 cells were infected at MOI8 and selected using 1 μg/ml puromycin for 24 hr. To further select for high-expressing cells, target and control shRNA-carrying cultures were treated for 18 hr with 1 μg/ml doxycycline collected in a single cell solution using Gentle Cell Dissociation Reagent (STEMCELL Technologies #07174) for 10 min at room temperature and sorted on a BD FACSAria II to obtain the top 20% of RFP- expressing cells. Doxycycline was immediately removed thereafter, and the cells were cultured for in the presence of the ROCK inhibitor Y-27632 (STEMCELL Technologies, #72304) for 48 hr to promote survival.

## Knockdown of SMARCB1 in steady state hESCs

For all KD experiments using steady state hESCs, high shRNA-expressing cells were split at a density of 1:24 and allowed to recover for 48 hr prior to the initiation of 1 µg/ml of doxycycline treatment. Treatment was continued for 3 days prior to collection for KD validation, RNAseq, or ATACseq. RNA was isolated using Norgen Total RNA Purification Plus kits (#48300), and for all qPCR experiments, cDNA was generated using an iScript cDNA Synthesis Kit (BioRad, #1708891). The primers used for qPCR analysis are available in *Supplementary file 6*.

## Monolayer and embryoid body neural induction protocol

Cells carrying inducible shRNAs against SMARCB1 were cultured for 3 days in the presence or absence of 1 µg/ml doxycycline. Cells were then collected in Gentle Cell Dissociation Reagent for 10 min at 37°C and dissociated into single cells. The remainder of the protocol was performed as per the STEMCELL Technology instructions for neural induction using STEMDiff Neural Induction Media (#05835), with minor modifications. Specifically, cells were plated at a density of $1.5 \times 10^5$ cells/cm$^2$, as this density provided the highest efficiency of induction. At 6 days, cells were collected for RNA-seq, ATACseq, or processed for immunohistochemistry. The monolayer protocol was performed three times with three biological replicates each, with similar results for each performance.

EBs were formed from control and SMARCB1 KD cells (following 3 days of dox treatment). The EBs were 1,000 cells each and were generated in AggreWell400 plates (StemCell Technologies #34421). After 24 hr, the EBs were transferred to ultra-low-adherence dishes (Corning #3471), following the StemCell Technologies protocol. The EBs were cultured in AggreWell EB Formation Medium (StemCell Technologies, #05893), with media replaced daily by allowing the EBs to settle in 15 ml conical tubes for 10 min prior to media aspiration. On Day four following EB formation, 1 µM all-trans retinoic acid was added to the media, whereupon the EBs were cultured for an additional 4 days prior to collection for isolation of RNA. The EB experiment was performed 5 times with 2–3 replicates per control/SMARCB1 KD condition for each experiment. Similar morphological effects of SMARCB1 KD were observed for each performance, and qPCR analysis was performed on replicates from one performance.

For both monolayer and EB experiments, RNA was isolated using Qiagen RNAeasy kits (#74104), and cDNA was generated using an iScript cDNA Synthesis Kit (BioRad, #1708891). The primers used for the analysis of neural and pluripotency markers are available in *Supplementary file 6*.

## Definitive endoderm differentiation protocol

Cells carrying inducible shRNAs against SMARCB1 were cultured for 3 days in the presence or absence of 1 µg/ml doxycycline. Cells were then collected in Gentle Cell Dissociation Reagent for 10 min at 37°C and dissociated into single cells. The remainder of the protocol was performed as per the STEMCELL Technology instructions for endodermal differentiation using STEMDiff Definitive Endoderm Kit (#05110). The experiment was performed three times, with results being tested by qPCR (n = 3 biological replicates/condition), flow cytometry (total n = 5 biological replicates/condition), and/or immunohistochemistry (n = 1 biological replicate/condition).

## Mesodermal induction

Cells carrying inducible shRNAs against SMARCB1 were cultured for 4 days in the presence or absence of 1 µg/ml doxycycline until ≈90% confluency. Cells were then cultured for 2 days in RPMI1640 supplemented with NeuroCult without insulin (StemCell Technologies #05733) and 5 µM CHIR99021 (GSK inhibitor, StemCell Technologies, #72052), after which they were treated for two additional days in RPMI1640 supplemented with NeuroCult without insulin. The mesodermal induction experiment was performed twice, with qPCR results being based on three biological replicates per condition and immunohistochemistry results being based on one biological replicate per condition.

## Immunohistochemistry

Cells subjected to the neural induction protocol were processed for IHC as described it the STEMdiff Human Neural Progenitor Antibody Panel (StemCell Technologies, #69001). Briefly, cells in glass-bottom plates were fixed for 15 min at room temperature in 4% formaldehyde, permeabilized with

0.1% Tween for 10 min, blocked with 5% FBS in PBS, and stained with antibodies against PAX6 (anti-rabbit, StemCell Technologies #60094 (1:500, Lot# SC09342), or BioLegend #901301, (1:300, Lot# B235967), NESTIN (anti-mouse, StemCell Technologies #60091 (1:1000, Lot# SC09341), and OCT4 (anti-mouse, StemCell Technologies #60093, 1:1000, Lot# SC09338). After 3x rinses with PBS, the cells were incubated with secondary antibodies for 1 hr at room temperature (secondaries available upon request) and counterstained with DAPI (ProLong Diamond Antifade Mountant, Thermo-Fisher #P36971). Images were obtained on a Zeiss LSM 710 inverted confocal microscope and analyzed in ImageJ (*Schneider et al., 2012*). Endodermal and mesodermal differentiation cultures were processed in the same manner and probed with antibodies against SOX17 (AF1924, R and D Systems, 1:500) and EOMES (MAB6166, R and D Systems, 10 µg/ml, Lot# CEDQ0218011), respectively.

## Western blotting

For detection of SWI/SNF subunit protein levels, cells were lysed in Buffer X (100 mM Tris-HCL, pH 8.5, 250 mM, 1% NP-40, 1 mM EDTA) containing 1:100 protease inhibitor cocktail (Pierce Technology, #PI78442) and 1:100 PMSF on ice for 20 min. The lysates were then homogenized by vortexing for 30 s and centrifuged for 12,800 x g for 15 min. The protein concentration in the supernatant was quantified using the Bradford assay, and 30–50 µg protein was separated on 4–12% Tris-glycine gels (ThermoFisher, XP04122BOX) at 100 V for 2 hr. The gels were then transferred to PVDF membranes for 2 hr at 400 mA at 4 °C. The blots were then blocked for 1 hr at room temperature or 4° overnight in TBS containing 5% milk. The membranes were then incubated overnight at 4° with the following primary antibodies diluted in TBS-Tween containing 5% milk: SMARCA4 (lab-generated antibody targeting aa437–678, anti-rabbit, 1:2000 [*Wade et al., 2015*] #133)), SMARCC1 (H76, Santa-Cruz, anti-rabbit, 1:500), SMARCC2 (E-6, Santa-Cruz, anti-mouse, 1:200), SMARCD1 (23, Santa-Cruz, anti-mouse 1:1000), SMARCE1 (lab-generated, anti-rabbit, 1:2000 [*Chen and Archer, 2005*]), Lamin A/C (H-110, Santa-Cruz, anti-rabbit), GAPDH (6C5, anti-mouse, 1:10,000, Abcam). Secondary antibody staining was performed with Li-Cor IRDye 800CW and IRDye 680RD antibodies against the appropriate species. Image acquisition was performed using an Odyssey CLx Infrared Imaging System and analyzed using ImageStudio Lite software.

## Flow cytometry

H1 hESCs subjected to the endodermal differentiation protocol were washed with PBS and treated with trypsin for 3 min. The cells were spun at 300xg for 5 min, resuspended in TeSR-E8, and stained with BV421 Mouse Anti-Human CD184 (BD Biosciences, cat# 566282), as per the manufacturer's instructions. Cells were then re-pelleted and washed 2x with TeSR-E8 and assayed on a Becton Dickinson LSR II Flow Cytometer (BD Biosciences) or a Becton Dickinson LSRFortessa following an additional stain with propidium iodide as a vital dye. Only live cells were considered for the analysis, with data being analyzed using FACSDiva software v8.0.2.

## RNAseq

Three replicates were used for all RNAseq experiments. RNA was isolated from H1 hESCs and neural/induction cells using Norgen Total RNA Purification Plus kits (#48300), and RNAseq with rRNA removal was performed by Expression Analysis, an IQVIA company. The raw data were filtered for quality using sickle (default parameters), adapters were removed, and reads were aligned to hg19 using STAR, keeping only unique alignments. Feature counts were obtained using bedTools feature-Counts (steady state hESCs) or Salmon (neural induction). The R package *limma* was used to call regions with differential read counts among genes/regions with a max group mean (mgm) of 8 (hESCs) or 32 (neural induction).

## ATACseq

Two replicates were used for all ATAC experiments. Cells were dissociated by incubation with Gentle Cell Dissociation Reagent (steady state hESCs) or Accutase (neural induction) (StemCell Technologies, #07920) for 10 min at 37°C. Cells were then collected by pipetting, counted, and spun for 5 min at 300 x g, after which they were resuspended in to 1e6 cells/ml. Cells were then subjected to the ATACseq protocol as described by *Buenrostro et al. (2015)*, with the following parameters.

Buffer: CSK buffer (10 mM PIPES pH 6.8, 100 mM NaCl, 300 mM sucrose, 3 mM MgCl2, 0.1% Triton X-100). Transposase volume/25 µl cell suspension: 5 µl. Transposase treatment time: 30 min, with mixing every 10 min. Libraries were sequenced at the NIEHS Epigenomics Core Facility on the Next-Seq 500 platform, and reads were trimmed using default parameters (steady state hESC experiments) or with -q 26 (neural induction experiment), with the latter modification made due to a technical issue with the sequencing run. Reads were aligned to hg19 with bowtie2 with the following parameters: -X 2000 N 1 –no-unal –no-mixed –dovetail –no-discordant. Uniquely aligned reads were filtered using an in-house python script, which was also used to remove mitochondrial reads. Reads $\leq$ 135 bp were retained for analysis of accessible regions and deduplicated with picard, using the following parameters: MAX_RECORDS_IN_RAM = 5000000, REMOVE_DUPLICATES = true. Peaks were called for each replicate using macs2, with the following parameters: callpeak –nomodel –nolambda –keep-dup all –slocal 10000 -q 0.005 [https://github.com/taoliu/MACS]. After the top 0.1% of Regions of Exceptionally High Depth of Aligned Short Reads were removed from the called peak files, Untreated, Day 2, and Day 3 KD peaks were merged and used as the considered regions for statistical analysis (*Pickrell et al., 2011*). Differential peaks were called using the R package *limma*, with cut-offs of q < 0.01, log2(FC) >2, and a max group mean (mgm) of $\geq$32 reads. Peaks in which this mgm criterion was not met in any condition were excluded from further analyses.

The signal normalization factors required to permit comparison of conditions were determined by finding the scale factors required to equalize the median read counts for all considered regions of interest. For these determinations, to exclude the use of regions with signals that could not be distinguished from noise, only regions with signals above the median signal value were considered.

## Differential peak/gene significance analysis, enrichment analyses

For each of the differential ATAC-seq peak sets, ranges of increasing size were made around the peaks (from 5 kb to 2 Mb beginning with 5 kb increments). Analyses were performed using R version 3.5.1 and visualized with ggplot2 version 3.1.0 (*R Core Team, 2018*; *Wickham, 2016*). Regions and overlaps were determined using Bioconductor 2.42.0 packages rtracklayer 1.42.1 and GenomicRanges 1.34.0 (*Huber et al., 2015*; *Lawrence et al., 2009*; *Lawrence et al., 2013*). Ranges were intersected with the TSS of all RNA-seq genes in the given analysis (i.e., steady state or neural induction), and a hypergeometric test was used to determine if differential RNA-seq TSS gene hits were enriched in the regions overlapping the differential ATAC-seq peaks. In addition, for each analysis, an equivalent random subset of observed ATAC-seq peaks were selected 1000 times and tested for enrichment using the same procedure. The data were collected to visualize the P-value of observed differential ATAC-seq peaks, the median and 5–95% quantile range of random ATAC-seq peaks. The detailed workflow is available in a supplementary file (Peak_gene_enrichment_analysis.html).

To assess the significance of overlaps between differentially accessible peaks and genomic regions of interest, the following command was performed in R: phyper(q, m, n, k, lower.tail = FALSE), where q = the number of intersection between the differential peak set and the genomic regions of interest, m = the intersection between all considered ATAC peaks, regardless of condition, and the genomic regions of interest, n = the number of all considered ATAC peaks not intersecting the genomic regions of interest, and k = the number of differential peaks in the considered set.

## HOMER

HOMER motif analysis was used with findMotifsGenome.pl, using the following parameters: -size given, background regions defined as non-differential peaks that were similar to the considered differential peak set in terms of size, ATAC signal, and distance to the nearest promoter (*Heinz et al., 2010*). Motifs were considered statistically enriched at p<1E-50.

## qPCR

All qPCR results were calculated using the ΔΔCt method relative to control cell populations, as indicated in the figure legends, and were normalized to the geometric mean of *18S* and *GAPDH* levels, with 3–4 technical replicates per sample. The bar heights represent mean relative expression for three biological replicates, and error bars represent standard deviations.

## Data availability

All raw RNAseq and ATACseq data have been made available in NCBI's Gene Expression Omnibus (*Edgar et al., 2002*), with accession number GSE128351.

## Other software tools

Several of the described bioinformatic analyses were performed using samtools: (v0.1.20), bedTools (v2.21.0), deepTools (v2.5.3), Picard (v2.9.2), GNU Parallel (v20170522), and RStudio (v1.1.383). Pathway analysis was performed using Ingenuity Pathway Analysis: v 01–10. Flow cytometry data were acquired and analyzed using BD FACSDiva: 8.0.2.

# Acknowledgements

The authors would like to thank the following NIEHS facilities and groups for their assistance in data acquisition and processing: The Epigenomics Core Laboratory, the NIEHS Integrative Bioinformatics Group, the Fluorescence Microscopy Imaging Center, the Flow Cytometry Core, the Office of the Director – Illustration Services, and the Viral Vector Core Laboratory. We would also like to thank Drs. Guang Hu, Paul Wade, Serena Dudek, David Fargo, and Raja Jothi, for critical reading of the manuscript and advice. Drs. Jackson Hoffman, Eric Milliman, and Nolan Gokey are thanked for their advice on experimental procedures and analyses. LFL is supported by a Postdoctoral Research Associate (PRAT) fellowship from the National Institute of General Medical Sciences (NIGMS), project number GM120018. This research was supported by the Intramural Research Program of the NIH – National Institute of Environmental Health Sciences [Z01 ES071006-18; nih.gov].

# Additional information

### Funding

| Funder | Grant reference number | Author |
| --- | --- | --- |
| National Institute of Environmental Health Sciences | Z01 ES071006-18 | Trevor K Archer |
| National Institute of General Medical Sciences | GM120018 | Lee F Langer |

The funders had no role in study design, data collection and interpretation, or the decision to submit the work for publication.

### Author contributions

Lee F Langer, Conceptualization, Data curation, Formal analysis, Validation, Investigation, Methodology, Writing—original draft, Writing—review and editing; James M Ward, Data curation, Formal analysis; Trevor K Archer, Conceptualization, Resources, Supervision, Funding acquisition, Investigation, Project administration, Writing—review and editing

### Author ORCIDs

Lee F Langer (iD) https://orcid.org/0000-0002-9819-475X
Trevor K Archer (iD) https://orcid.org/0000-0001-7651-3644

### Decision letter and Author response

Decision letter https://doi.org/10.7554/eLife.45672.024
Author response https://doi.org/10.7554/eLife.45672.025

# Additional files

### Supplementary files

• Supplementary file 1. Compilation of shRNAs, differentially regulated genes in SMARCB1 KD cells and the overlap between their TSSs and publicly available genomic features in hESCs. Sheet 1, The

shRNAs used in the present study. Sheet 2, The most strongly upregulated and downregulated genes in *SMARCB1* KD cells based on RNAseq analysis. Sheet 3, The degree and significance of the overlap between the TSS of differentially regulated genes and publicaly available ChIPseq datasets for for transcription factors and histone marks in hESCs.
DOI: https://doi.org/10.7554/eLife.45672.013

• Supplementary file 2. Comparison accessibility peaks in SMARCB1 KD cells and publicly available genomic datasets in hESCs, along with Motif analysis of accessibility peaks. Sheets 1–2, Degree and significance of overlap between lower and higher accessibility peaks in *SMARCB1* KD cells and publicly available ChIPseq datasets for transcription factors and histone marks in hESCs. Sheets 3–4, Motif analysis using HOMER of lower and higher accessibility peaks in *SMARCB1* KD cells.
DOI: https://doi.org/10.7554/eLife.45672.014

• Supplementary file 3. Top differentially affected genes in control in SMARCB1 KD cells subjected to neural induction.
DOI: https://doi.org/10.7554/eLife.45672.015

• Supplementary file 4. Differentially expressed genes (DEGs) with promoterscontaining differentially accessible peaks and discovered IPA categories in nervous system development and function; along with DEGs in the SMARCB1KD group within 500 kB of an LA peak. Sheet 1: Differentially expressed genes (q < 0.05, FC > 1.5) with promoters containing differenitally accessible peaks (q < 0.01, FC > 2). Sheet 2: List of IPA categories in Nervous System Development and Function with associated p-values of enrichment and activation scores when considering genes within 500 kb of lower accessibility peaks in SMARCB1KD cells following the neural induction protocol. The category Development of Neurons is highlighted in yellow. Sheet 3: The genes within this category that show significantly different expression in the SMARCB1 KD group and that fall within 500 kB of an LA peak.
DOI: https://doi.org/10.7554/eLife.45672.016

• Supplementary file 5. Superenhancers with higher expression levels, and those that are near genes with differential expression levels. Sheet 1: Column A: Tissue/cell line; Column B: Number of defined super-enhancers in tissue/cell line; Column C: Number of all considered accessible regions that overlap super-enhancers; Column D: Percentage of all considered accessible regions that overlap super-enhancers; E: Number of lower accessibility regions that overlap super-enhancers; F: Percentage of lower accessibility regions that overlap super-enhancers; G: p-value of overlap between super-enhancers and lower accessibility regions; H: Number of higher accessibility regions that overlap super-enhancers; I: Percentage of higher accessibility regions that overlap super-enhancers; J: p-value of overlap between super-enhancers and higher accessibility regions. Blue cells indicate significant overlaps with lower accessibility regions, and red cells indicate significant overlaps with higher accessibility regions. Sheet 2: Columns A-C: Location of superenhancer; Column B: q-value of change in super-enhancer RNA expression; Column D: Log2 fold difference between in super-enhancer RNA expression between SMARCB1KD and control cells following the neural induction protocol. Columns E-F: The number of RNAseq reads aligned to the super-enhancer in the control and SMARCB1 KD conditions, respectively. Sheet 3: Column A: Name of differentially expressed gene by RNAseq; Column B: q-value of change in gene expression; Column C: Log2 fold difference in gene expression between SMARCB1KD and control cells following the neural induction protocol. Columns D-F: The location of the super-enhancer with differential eRNA expression. G: Distance from the super-enhancer with differential eRNA expression to the TSS of the gene in column A. Sheet 4: Significant overlap between higher accessibility regions following the neural induction protocol in SMARCB1 KD cells and hESC ChIPseq peaks. The differential accessibility peaks in SMARCB1 KD cells following the neural induction protocol were intersected with available ChIPseq peaks for hESCs. The degree of intersection was assessed for significance using hypergeometric tests. The transcription factor ChIPseq peaks that significantly represented (p<0.05) in higher accessibility peaks are highlighted in in red. No ChIPseq peak set significantly overlapped with lower accessibility peaks.
DOI: https://doi.org/10.7554/eLife.45672.017

• Supplementary file 6. The qPCR primers used in this study.
DOI: https://doi.org/10.7554/eLife.45672.018

• Transparent reporting form
DOI: https://doi.org/10.7554/eLife.45672.019

## Data availability

All raw RNAseq and ATACseq data have been made available in NCBI's Gene Expression Omnibus (Edgar, 2002), with accession number GSE128351.

The following dataset was generated:

| Author(s) | Year | Dataset title | Dataset URL | Database and Identifier |
|---|---|---|---|---|
| Langer LF | 2019 | Tumor suppressor SMARCB1 suppresses super-enhancers to govern hESC 2 lineage determination | http://www.ncbi.nlm.nih.gov/geo/query/acc.cgi?acc=GSE128351 | NCBI Gene Expression Omnibus, GSE128351 |

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
