## [Decision Letter]

Thank you for submitting your article "Tumor suppressor SMARCB1 suppresses super-enhancers to govern hESC lineage determination" for consideration by *eLife*. Your article has been reviewed by Jessica Tyler as the Senior Editor, a Reviewing Editor, and two reviewers. The reviewers have opted to remain anonymous.

The reviewers have discussed the reviews with one another and the Reviewing Editor has drafted this decision to help you prepare a revised submission.

Summary:

This study addresses the role of the SMARCB1 subunit of the ATP-dependent SWI/SNF chromatin remodeling complex in regulating gene expression in human embryonic stem cells (hESCs) before and after differentiation. Study of SMARCB1 is particularly important, because mutations in this subunit are strongly associated with pediatric neural tumors. The authors compare gene expression (RNA-seq) and chromatin accessibility (ATAC-seq) in control and SMARCB1 knockdown (KD) hESCs made using a lentivirus carrying a dox-inducible shRNA. The effects of initiating differentiation into neural cells, endoderm or mesoderm are also assessed using the same methods.

In undifferentiated hESCs, many genes are activated in response to knockdown, whereas only a few are repressed, indicating that SMARCB1 has a primarily repressive role. This observation is particularly true of genes with bivalent chromatin marks. Changes in chromatin accessibility occur primarily at enhancers: the accessibility of active enhancers decreases in KD cells (LA peaks), indicating that SMARCB1 promotes accessibility at active enhancers. In contrast, super-enhancers show the opposite trend, indicating that SMARCB1 represses their function (HA peaks). They find that SMARCB1 KD hESCs can be differentiated into mesoderm or endoderm but not neural cells. In the latter case, only some neural genes are induced and many genes characteristic of stem cells remain active. They conclude that SMARCB1 is needed for the initial stages of neural induction, but not for mesoderm or endoderm induction. In particular, the hESC super-enhancers remain accessible in neurally induced KD cells.

Overall, the reviewers found the study to be interesting and felt that the major claims were supported by the presented results. The reviewers did express some difficulty in following the text in places and requested that more explanation of the analyses be included in the text.

Essential revisions:

1) In Figure 2B, the aim appears to be to correlate changes in ATAC-peaks with changes in gene expression. This is tricky because the target genes of specific enhancers are unknown and difficult to infer because they are often separated by large genomic distances. The reviewer did not understand the method used in Figure 2B to calculate the probability of association – what assumptions underlie this analysis? Detailed explanation in the text is needed.

2) The reviewer also found the "dot plots" such as those in Figure 2C/D difficult to understand. In Figure 2C, I think they have calculated the probability of low (LA) and high (HA) accessibility ATAC-peaks coinciding with enhancers and super-enhancers found in 98 human cells using data from other papers (subsection “*SMARCB1* KD differentially affects chromatin accessibility at key regions associated with hESC identity”). Some significant dots in Figure 2C are labeled – but what cell type do they belong to? Presumably the current experiments. The reviewer was not sure why they did this analysis. Isn't it sufficient to examine the correlations between enhancer types and LA/HA peaks in the cells used here? The justification for this analysis seems obscure, but perhaps I'm missing something. More explanation is needed in the text.

3) What about the promoters of differentially expressed genes? Do their accessibilities change as expected from the RNA-seq data?

4) Figure 1F shows the significance of overlap between up- or down-regulated genes and the genomic binding sites of numerous regulatory proteins or histone modifications. In this figure, it would be reasonable to include the binding sites of any SWI/SNF subunits for which ChIP-seq data are available to better assess the extent to which the gene expression changes are direct.

5) In hESCs, many more genes go up than down in SMARCB1 KD cells, suggesting SMARCB1 largely represses (or at least limits) gene expression. On the other hand, many more regions become less accessible than more accessible in SMARCB1 KD hESCs, which suggests a greater role in activation of transcription. How do the authors explain these phenotypes?

6) Figure 2B, Figure 4D, and Figure 5C seem strange in that no matter how large the bin size surrounding a region of interest (e.g., differential ATAC peaks), the association with differentially expressed genes is statistically significant. For lower accessibility peaks in Figure 2B, significance actually increases as the bin size increases from 300 to 500 kb from the aggregate ATAC-seq peak center. This seems unlikely. Furthermore, Figures 4D and Figure 5C seem to flatten out on the right as bin size gets very large, but they seem to approach an asymptote of -log10(p) = 2-3 (e.g., p = 0.01-0.001). Shouldn't they approach -log10(p) = 0? How might a set of "peaks" consisting of random locations in the genome (with the number of different random locations equal to the number of real peaks) look when subjected to the same analysis? This comparison could be added to these figures.

7) Subsection “SMARCB1 is required for the initial stages of neural induction but is dispensable for endodermal and mesodermal induction”: "In contrast, some of the genes with the highest expression in SMARCB1 KD vs. control cells included the neural differentiation markers FOXG1 (FD = -28.2), DLK (FD = -25.9), and LHX2 (FD = -14.2) (Supplementary file 3) (Watanabe et al., 2005, Porter, 1997, Tedeschi and Bradke, 2013)." Here, I think "highest" should be either "lowest" or "most strongly repressed".

---

## [Author Response]

Essential revisions:1) In Figure 2B, the aim appears to be to correlate changes in ATAC-peaks with changes in gene expression. This is tricky because the target genes of specific enhancers are unknown and difficult to infer because they are often separated by large genomic distances. The reviewer did not understand the method used in Figure 2B to calculate the probability of association – what assumptions underlie this analysis? Detailed explanation in the text is needed.

The reviewers are correct in that attributing changes at a particular enhancer to a particular gene is not possible given the available data. For this reason, we chose instead to infer such effects at a bulk level by considering enriched proximity of differential ATAC peaks to differentially expressed genes (DEGs) as indirect evidence of peak/gene interactions. Specifically, the number of DEGs within a given range of all differentially accessible ATAC peaks was compared to the total number of TSS within those ranges. A hypergeometric test was then used to assess whether more DEGs fell within this range than would be expected by chance. For example, when ranges of 20 kb in either direction were made around all HA peaks in the steady state condition, 955 TSS are encompassed, 132 of which are upregulated following *SMARCB1* KD. These values respectively correspond to 5.3% of the total gene set and 7.5% of the upregulated gene set, indicating a significant enrichment of upregulated genes, with a p-value of 7.6e-5. These values were then plotted for all ranges out to 2 MB (Figure 2B) in the revised manuscript. A visual workflow was added to Figure 2—figure supplement 1C, and the following explanation and example were added to subsection “SMARCB1 protein reduction relieves repression of bivalent genes in hESCs”. Additional detail was also added to the Methods, and as HTML file with a detailed computational workflow is provided in the supplementary material.

“An important readout of modified chromatin accessibility is differential gene expression, although it is difficult to assign changes in a particular ATAC peak to changes in transcriptional output. We therefore utilized an approach in which the number of differentially expressed genes (DEGs) within a given distance of all differentially accessible ATAC peaks was compared to the total number of genes within those ranges. A hypergeometric test was then used to assess whether more DEGs fell within this range than would be expected by chance. For example, when ranges of 20 kb were made around all HA peaks, 955 TSS are encompassed, 132 of which were upregulated following *SMARCB1* KD. These values respectively correspond to 5.3% of the total gene set and 7.5% of the upregulated gene set, indicating a significant enrichment of upregulated genes, with a p-value of 7.6e-5 (Also see Figure 2—figure supplement 1C).”

2) The reviewer also found the "dot plots" such as those in Figure 2C/D difficult to understand. In Figure 2C, I think they have calculated the probability of low (LA) and high (HA) accessibility ATAC-peaks coinciding with enhancers and super-enhancers found in 98 human cells using data from other papers (subsection “SMARCB1 KD differentially affects chromatin accessibility at key regions associated with hESC identity”). Some significant dots in Figure 2C are labeled – but what cell type do they belong to? Presumably the current experiments. The reviewer was not sure why they did this analysis. Isn't it sufficient to examine the correlations between enhancer types and LA/HA peaks in the cells used here? The justification for this analysis seems obscure, but perhaps I'm missing something. More explanation is needed in the text.

The reviewer is correct in that the enhancer sets labeled in Figure 2C are from the cell types used in the present analysis, as are the data in Figure 2E; the figure and text have been modified appropriately. The unlabeled dots in Figure 2C reflect all published super enhancer data sets for other cell types. The reasoning behind the inclusion of these data is that manipulation of the SWI/SNF complex has been shown to promote or repress certain differentiation pathways (Zhang et al., 2014, Wade et al., 2015). As super-enhancers are key regions of cell identity, we sought to assess whether *SMARCB1* KD would preferentially alter accessibility dynamics within particular super-enhancer sets and thereby illuminate any effects of this manipulation on hESC differentiation (Hnisz et al., 2013). This explanation has been added to the text of the applicable Results section:

“The super-enhancers of other cell types were included in this analysis given previous results indicating that manipulation of the SWI/SNF complex can promote or repress particular fates (Wade et al., 2015, Zhang et al., 2014). As super-enhancers are key regions associated with cell identity, the preferential localization of differential ATAC peaks in these regions for other cell types would inform any effects of *SMARCB1* KD on differentiation.”

3) What about the promoters of differentially expressed genes? Do their accessibilities change as expected from the RNA-seq data?

In the neural induction experiments, multiple differentially expressed pluripotency-related genes contained higher accessibility peaks (in the *SMARCB1* KD condition) in their promoters, including *OCT4, mir302, DNMT3B, ZIC3*, and *DPPA2/4*. Similarly, multiple neural/neural induction-related genes that were expressed at lower levels in the *SMARCB1* KD condition, including *CER1, CRX, DPP6, SIX1*, contained lower accessibility peaks in their promoters. These data were added to Supplementary Table 4.

The situation differed in the steady state condition, where fewer than 5% of differentially accessible peaks were near promoters, and very few differential peaks were observed in the promoter regions of differentially expressed genes (N=0 higher accessibility peaks in promoters of upregulated genes, N=6 lower accessibility peaks in the promoters of downregulated genes). The accessibility at the promoters of upregulated genes do show a modest increase in the *SMARCB1* KD condition, and the signal at downregulated gene promoters shows little change (see Author response image 1). However, the lack of statistically significant accessibility changes in these regions indicate that the effects of differential accessibility on transcriptional output are mediated principally distal to the target gene. This result was not unexpected as promoter accessibility correlates rather poorly with gene expression in pluripotent cells. This is because the promoters of developmental genes remain poised and strongly accessible, even though they may be expressed at low levels in ESCs (Schlesinger and Meshorer, 2019). For example, Author response image 1 is a browser track of accessibility data for 4 kB around the *PAX6* promoter in steady state hESC (top) and neural induction conditions. The signals are very similar despite the >500-fold expression difference in *PAX6* between the conditions. We believe that the differential peak vs. differential gene data graphed in Figure 2B convincingly demonstrate that the effects of *SMARCB1* KD on accessibility are mediated distally to the TSS.

4) Figure 1F shows the significance of overlap between up- or down-regulated genes and the genomic binding sites of numerous regulatory proteins or histone modifications. In this figure, it would be reasonable to include the binding sites of any SWI/SNF subunits for which ChIP-seq data are available to better assess the extent to which the gene expression changes are direct.

We agree with the reviewers that determining the degree of intersection between differential gene promoter and SWI/SNF factor binding sites would is appropriate. In fact, the publicly available ChIPseq data for BRG1 (Rada-Iglesias et al., 2011) was included in this analysis, although the results did not reach significance. However, BRG1 peaks are biologically enriched at most hESC promoters. Specifically, BRG1 peaks were observed in 16,031 of the 17,462 genes (92%) considered in the RNAseq analysis. In contrast, BRG1 peaks were present at 90% of the promoters of upregulated genes following *SMARCB1* KD and 88% of the promoters of downregulated genes. Although significance is not reached for these comparisons, the high frequency of BRG1 binding at the promoters of differentially expressed genes clearly indicates that the SWI/SNF is active at these promoters. The text has been modified to include this information:

“We also tested for enrichment for the only SWI/SNF subunit for which ChIPseq data are available in this cell type, the catalytic subunit SMARCA4. Although these peaks did not emerge as significantly enriched near the TSS of differentially expressed genes (DEGs), this is largely due to SMARCA4 being highly biologically enriched at most promoters in hESCs. Specifically, SMARCA4 peaks are present at 16,031/17,462 (92%) of the considered genes in the RNAseq analysis, whereas these values were 90% and 88% for up- and downregulated genes, respectively (Supplementary file 1) (Rada-Iglesias et al., 2011).”

5) In hESCs, many more genes go up than down in SMARCB1 KD cells, suggesting SMARCB1 largely represses (or at least limits) gene expression. On the other hand, many more regions become less accessible than more accessible in SMARCB1 KD hESCs, which suggests a greater role in activation of transcription. How do the authors explain these phenotypes?

We have also noticed this difference in the effects of *SMARCB1* KD on accessibility and transcription and note, as mentioned in the response to comment #3, that accessibility at promoters is not a strong prediction of transcriptional output in ESCs. We also hypothesize that SMARCB1 positively affects transcription by other mechanisms than altered accessibility. For example, SMARCB1 may be required for the recruitment of transcription factors or transcriptional machinery to promoters hESCs, which would result in decreased transcription but not a dramatic change in accessibility. Evidence for transcriptional control independent of accessibility changes include the established antagonism between the SWI/SNF and PRC2 complexes, the interaction between the complex and the tumor suppressor p53, and the observed associations between SMARCB1 and RNA Pol I and RNA Pol II (Kadoch et al., 2017, Lee, 2002, Cho, 1998, Zhai et al., 2012).” This hypothesis has been added to the Discussion section:

“It is also worth noting that while there were a greater number of lower accessibility peaks in the steady *SMARCB1* KD condition, there was a strong bias towards transcriptional upregulation. We hypothesize that SMARCB1 positively affects transcription by other mechanisms than altered accessibility. For example, SMARCB1 may be required for the recruitment of transcription factors or transcriptional machinery to promoters hESCs, which would result in decreased transcription but not a dramatic change in accessibility. Evidence for transcriptional control independent of accessibility changes include the established antagonism between the SWI/SNF and PRC2 complexes, the interaction between the complex and the tumor suppressor p53, and the observed associations between SMARCB1 and RNA Pol I and RNA Pol II (Kadoch et al., 2017, Lee, 2002, Cho, 1998, Zhai et al., 2012).”

6) Figure 2B, Figure 4D, and Figure 5C seem strange in that no matter how large the bin size surrounding a region of interest (e.g., differential ATAC peaks), the association with differentially expressed genes is statistically significant. For lower accessibility peaks in Figure 2B, significance actually increases as the bin size increases from 300 to 500 kb from the aggregate ATAC-seq peak center. This seems unlikely. Furthermore, Figure 4D and Figure 5C seem to flatten out on the right as bin size gets very large, but they seem to approach an asymptote of -log10(p) = 2-3 (e.g., p = 0.01-0.001). Shouldn't they approach -log10(p) = 0? How might a set of "peaks" consisting of random locations in the genome (with the number of different random locations equal to the number of real peaks) look when subjected to the same analysis? This comparison could be added to these figures.

We have performed the differential ATAC peak vs. differential gene association analysis out to 2 Mb for each of the mentioned subfigures, and we indeed observe that the p-value drops below significance (p=0.01) over distance in each case. These extended data have been substituted for Figure 2B, Figure 4D and Figure 5C, respectively. In addition, to confirm that the observed significance of the associations is specific to those ATAC peaks that are differentially accessible, we generated 1000x sets of random ATAC peaks that were matched in size and number to the differential peaks and subjected these sets to the same analysis. In each case, we found that these random peak sets displayed levels of significance that were well below those of the considered differential peak set. The 5-95% confidence levels of the significance for these random sets are displayed as shaded regions in Figure 2B, Figure 4D and Figure 5C and the text/figure legends have been modified accordingly. We thank the reviewer for raising this point, as we believe that it substantially strengthens the results.

7) Subsection “SMARCB1 is required for the initial stages of neural induction but is dispensable for endodermal and mesodermal induction”: "In contrast, some of the genes with the highest expression in SMARCB1 KD vs. control cells included the neural differentiation markers FOXG1 (FD = -28.2), DLK (FD = -25.9), and LHX2 (FD = -14.2) (Supplementary file 3) (Watanabe et al., 2005, Porter, 1997, Tedeschi and Bradke, 2013)." Here, I think "highest" should be either "lowest" or "most strongly repressed".

We thank the reviewers for bringing this error to our attention. The sentence has been edited accordingly.